# Advancing A(H5N1) influenza risk assessment in ferrets through comparative evaluation of airborne virus shedding patterns

Joanna A. Pulit-Penaloza ✉, Troy J. Kieran, Nicole Brock, Jessica A. Belser, Xiangjie Sun, Hui Zeng, Claudia Pappas, Juan A. De La Cruz, Yasuko Hatta, Han Di, C. Todd Davis, Terrence M. Tumpey & Taronna. R. Maines

Recent A(H5N1) zoonotic cases linked to poultry and cattle in North America highlight the urgent need to assess the pandemic potential of emerging strains. Using male ferrets, we evaluate two B3.13 and two D1.1 genotype A(H5N1) viruses isolated from humans and observe fatal disease and varying capacities for direct contact transmission. To enhance pandemic risk assessment, we conduct aerosol sampling using cyclone BC251 and water condensation capture-based SPOT samplers and perform comparative analyses to include additional A(H5N1), A(H9N2), A(H7N9), and A(H1N1)pdm09 strains with known transmissibility profiles. Although none of the A(H5N1) strains transmit via the air, B3.13 viruses are detected at significantly higher levels compared to D1.1 strains. Here we show strong correlations between viral loads in nasal washes, airborne virus shedding, and transmissibility in ferrets, highlighting the value of these metrics for identifying zoonotic influenza viruses that may be adapting toward increased transmission potential.

A(H5N1), a highly pathogenic avian influenza (HPAI) virus, has caused outbreaks in birds and occasional human infections since its emergence in East Asia in 1996. Through ongoing genetic mutation, the virus has diversified into multiple genotypes and clades. Clade 2.3.4.4b A(H5N1) virus was introduced into North America via migratory wild birds on multiple occasions and was first detected in the U.S. during December 2021[1]. By 2024, following reassortment with North American avian influenza viruses, two major genotypes, B3.13 and D1.1, had been identified in the U.S. Genotype B3.13 has been detected in domestic poultry, dairy cattle, cats, and other mammals, and was associated with the first U.S. human case linked to cattle exposure in Texas[2,3]. Genotype D1.1, initially identified in wild birds and poultry outbreaks, was later detected in Nevada and Arizona dairy herds in 2025, likely introduced through contact with infected wild birds[4]. Between 2024 and 2025, over 70 confirmed human infections with viruses from both genotypes have been reported in North America, primarily following exposure to infected poultry or cattle. While most cases were mild, characterized by conjunctivitis, cough, sore throat, fever, or fatigue, severe illness occasionally occurred, including a fatal

case involving a D1.1 genotype virus in Louisiana, raising concerns about its pandemic potential[5]. Genomic analyses indicate that neither genotype currently shows evidence of widespread human-adaptive mutations[6]. Nonetheless, the expanding host range and sporadic detection of mutations associated with reduced antiviral susceptibility highlight the risk that a pandemic-capable strain could emerge. These developments underscore the importance of sustained surveillance, genomic monitoring, and experimental studies to assess the pandemic potential of evolving A(H5N1) viruses[7].

Research using the ferret model is essential for evaluating the mammalian transmission potential of A(H5N1) viruses and informing pandemic risk assessments[8]. Ferrets are considered the most relevant small-animal model for influenza studies due to their respiratory tract anatomy and clinical symptoms that are similar to those of humans[9]. Historically, airborne transmission of A(H5N1) viruses in ferrets was not observed prior to the emergence of clade 2.3.4.4b viruses[10,11], heightening concern about currently circulating strains. Notably, two B3.13 genotype strains, isolated from dairy farm workers in Texas [A/Texas/37/2024 (TX/37)] and Michigan [A/Michigan/90/2024 (MI/90)]

Influenza Division, Centers for Disease Control and Prevention, Atlanta, GA, USA. ✉e-mail: xzy5@cdc.gov

early in 2024, were shown to transmit between ferrets via the airborne route[12–14]. However, D1.1 genotype viruses have not yet been evaluated in the ferret model, limiting our understanding of their transmission potential and the associated public health impact.

Refining our understanding of the ferret transmission model is essential for robust influenza risk assessment. A key gap is how the kinetics and magnitude of viral replication in the ferret respiratory tract affect the levels of virus released into the air, and how this correlates with transmissibility. While higher nasal wash titers often coincide with transmission events, they do not always reliably predict airborne transmissibility across strains[15]. Detection of infectious virus in the air may be a better predictor of transmission[16–18]. However, progress in measuring airborne influenza virus has been hampered by the lack of standardized sampling procedures. Currently, no sampler has been validated as a gold standard for influenza that both efficiently captures low levels of virus into a small, concentrated collection volume while preserving viral viability[19]. Consequently, previous studies in humans and animal models have more frequently focused on quantifying viral genomic RNA in air samples, rather than infectious virus. Nonetheless, with the use of new-generation samplers and improved protocols, growing evidence across multiple influenza strains points to a link between airborne virus levels and transmissibility profiles, supporting routine airborne virus quantification as a complement to existing risk assessment protocols[11,14,18,20]. Recent B3.13 A(H5N1) isolates from the 2024 Texas and Michigan human cases showed limited but measurable airborne transmission in ferrets. These findings were corroborated by the detection of infectious virus in the air collected from inoculated ferrets[12,14]. An independent study also demonstrated a direct link between detection of airborne A(H5N1) virus and observed transmissibility in the ferret model[11]. In contrast, ferrets inoculated with non-transmitting A(H5N1) strains shed little or no infectious virus into the air, suggesting that the lack of transmission reflected insufficient levels of infectious virus in the air. What remains unknown is whether D1.1 genotype viruses are shed into the air and transmit via air, and whether more recent B3.13 isolates are further adapting toward increased airborne shedding and more efficient airborne transmissibility than what was observed with earlier isolates of that genotype.

In this work, we assess the pathogenesis and transmissibility of two D1.1 and two B3.13 A(H5N1) viruses isolated from humans to better understand the evolutionary dynamics of these genotypes and to inform pandemic risk assessments. We show that the A(H5N1) viruses have the capacity to cause fatal disease, exhibit varying capacities for transmission in the presence of direct contact, but do not display an airborne-transmissible phenotype. To further examine the relationship between transmission outcomes and airborne virus shedding in the ferret model, we conduct side-by-side comparisons of a cyclone sampler (BC251) and a water condensation-based sampler (SPOT). While both samplers demonstrate high utility for studying influenza viruses in the air, the data indicate that the BC251 sampler has greater collection efficiency, whereas the SPOT sampler preserves infectious virus more efficiently. We then compare the levels of both airborne genomic RNA and airborne infectious virus shed by the D1.1 and B3.13 A(H5N1) viruses, along with a panel of additional viruses, including contemporary and historical A(H5N1) strains, as well as human seasonal A(H1N1)pdm09 and zoonotic A(H7N9) and A(H9N2) strains, and show that B3.13 viruses are detected in the air at significantly higher levels compared to D1.1 strains. Lastly, we show strong correlations between viral loads in nasal washes, airborne virus shedding, and transmissibility profiles, and employ logistic regression to model the probability of airborne transmission.

## Results

### Pathogenesis of genotype B3.13 and D1.1 A(H5N1) influenza viruses in the ferret model

Genotype B3.13 A(H5N1) viruses associated with outbreaks in dairy cows have been shown to replicate efficiently in the ferret respiratory tract leading, in most cases, to severe disease followed by systemic spread and death within a week[12–14,21]. In this study, we evaluated the pathogenesis of two B3.13 genotype viruses and two D1.1 genotype viruses isolated from humans between July and November of 2024. A/Washington/239/2024 (WA/239; genotype D1.1) and A/Colorado/137/2024 (CO/137; genotype B3.13) viruses were isolated from patients exposed to infected poultry, A/British Columbia/PHL-2032/2024 (BC/PHL; genotype D1.1) was isolated from a patient with an unknown exposure source, and A/California/147/2024 (CA/147; genotype B3.13) was isolated from a patient exposed to infected cows (Table 1)[5,22].

To evaluate the pathogenesis of these strains, ferrets were inoculated intranasally with 6 log$_{10}$ PFU of virus and assessed for clinical signs and symptoms of infection. Viral replication kinetics were monitored by collecting nasal wash and rectal swab samples. All the inoculated ferrets displayed rapid weight loss, fever, nasal discharge, and lethargy. Diarrhea was also frequently observed, and infectious virus was detected in the blood of most of the ferrets indicating systemic spread (Table 2, Supplementary Fig. 1a, b). Mean maximum peak titers in nasal washes ranged from 5.3 to 7.6 log$_{10}$ PFU/mL, with virus detected in rectal swabs from the majority of inoculated ferrets (Fig. 1a–f, Supplementary Fig. 2a-f). All animals developed severe

## Table 1 | Influenza viruses used in the study

| Virus | Name in the study | Subtype | Exposure source[a] | DCT[b] | RDT[b] | Group in the study[c] | Ref.[d] |
|---|---|---|---|---|---|---|---|
| A/Colorado/137/2024 | CO/137 | H5N1 | avian | 3/3 | 0/3 | Non-transmissible | This study |
| A/California/147/2024 | CA/147 | H5N1 | bovine | NT | 0/3 | Non-transmissible | This study |
| A/Washington/239/2024 | WA/239 | H5N1 | avian | 2/3 | 0/3 | Non-transmissible | This study |
| A/British Columbia/PHL-2032/2024 | BC/PHL | H5N1 | unknown | NT | 0/3 | Non-transmissible | This study |
| A/Vietnam/1203/2004 | VN/1203 | H5N1 | avian | 0/4 | NT | Non-transmissible | 34 |
| A/Michigan/90/2024 | MI/90 | H5N1 | bovine | 6/6 | 3/6 | Low-transmissible | 12 |
| A/Anhui/1/2013 | Anhui/1 | H7N9 | avian | 4/4 | 2/6 | Low-transmissible | 28 |
| A/Anhui-Lujiang/39/2018 | AL/39 | H9N2 | avian | 3/3 | 5/6 | High-transmissible | 29 |
| A/Nebraska/14/2019 | NE/14 | H1N1pdm09 | human | NT | 3/3 | High-transmissible | 30 |

[a]All viruses were isolated from humans, with the sources of exposure specified.

[b]DCT- direct contact transmission model; RDT-respiratory droplet transmission model. Number of contact ferrets with detectable virus and/or detectable antibodies to homologous virus in serum over the total number of contact ferrets. NT-not tested. Transmission in ferrets was determined by the detection of infectious virus and/or seroconversion.

[c]Viruses were categorized into groups based on their airborne transmission frequencies observed in this work or referenced studies: non-transmissible (0% transmission frequency), low-transmissible (33–50% transmission frequency), and high-transmissible (83–100% transmission frequency).

[d]DCT and RDT transmission frequency in ferrets is reported in this study, or in the indicated reference.

**Table 2 | Summary of the pathogenesis of genotype B3.13 and D1.1 A(H5N1) viruses in ferrets**

| Virus | Genotype | NW titer[a] | RS titer[a] | Weight loss (%)[b] | Temp increase (C°)[c] | Nasal disch[d] | Diarrh[d] | Viremia[d] | Mort.[e] |
|---|---|---|---|---|---|---|---|---|---|
| CO/137 | B3.13 | 6.8 (d3) | 3.5 (2/6) | 13.2 | 2.3 (d1) | 6/6 | 2/6 | 4/5 | 6/6 (d2-5) |
| CA/147 | B3.13 | 7.6 (d2) | 3.7 (3/3) | 16.1 | 1.7 (d2) | 3/3 | 3/3 | 3/3 | 3/3 (d4-5) |
| WA/239 | D1.1 | 5.3 (d3) | 3.2 (3/6) | 14.3 | 1.8 (d1) | 6/6 | 3/6 | 4/5 | 6/6 (d4-7) |
| BC/PHL | D1.1 | 6.7 (d1) | 3.7 (3/3) | 14.4 | 2.2 (d1) | 3/3 | 3/3 | 3/3 | 3/3 (d4-5) |

[a] Ferrets were intranasally inoculated with 6 log10 PFU of virus in 1 mL. Average maximum nasal wash (NW) titer; median peak day is in parenthesis. Average maximum rectal swab (RS) titer; number of ferrets with detectable virus over the total number of ferrets is in parenthesis. Titers are expressed as log10 PFU/mL; limit of detection is 10 PFU/mL.
[b] Average maximum weight loss on the day of final assessment.
[c] Average maximum temperature increase over the baseline (37.5-39.0 °C); median peak day is in parenthesis.
[d] Number of ferrets with nasal discharge, diarrhea or viremia over the total number of animals. One WA/239 and one CO/137 virus-inoculated animal succumbed to infection before blood sample collection and were therefore excluded from the total count.
[e] Number of animals that succumbed to infection or were euthanized during the experiment due to severe illness over the total number of animals, with the day range indicated in parentheses.

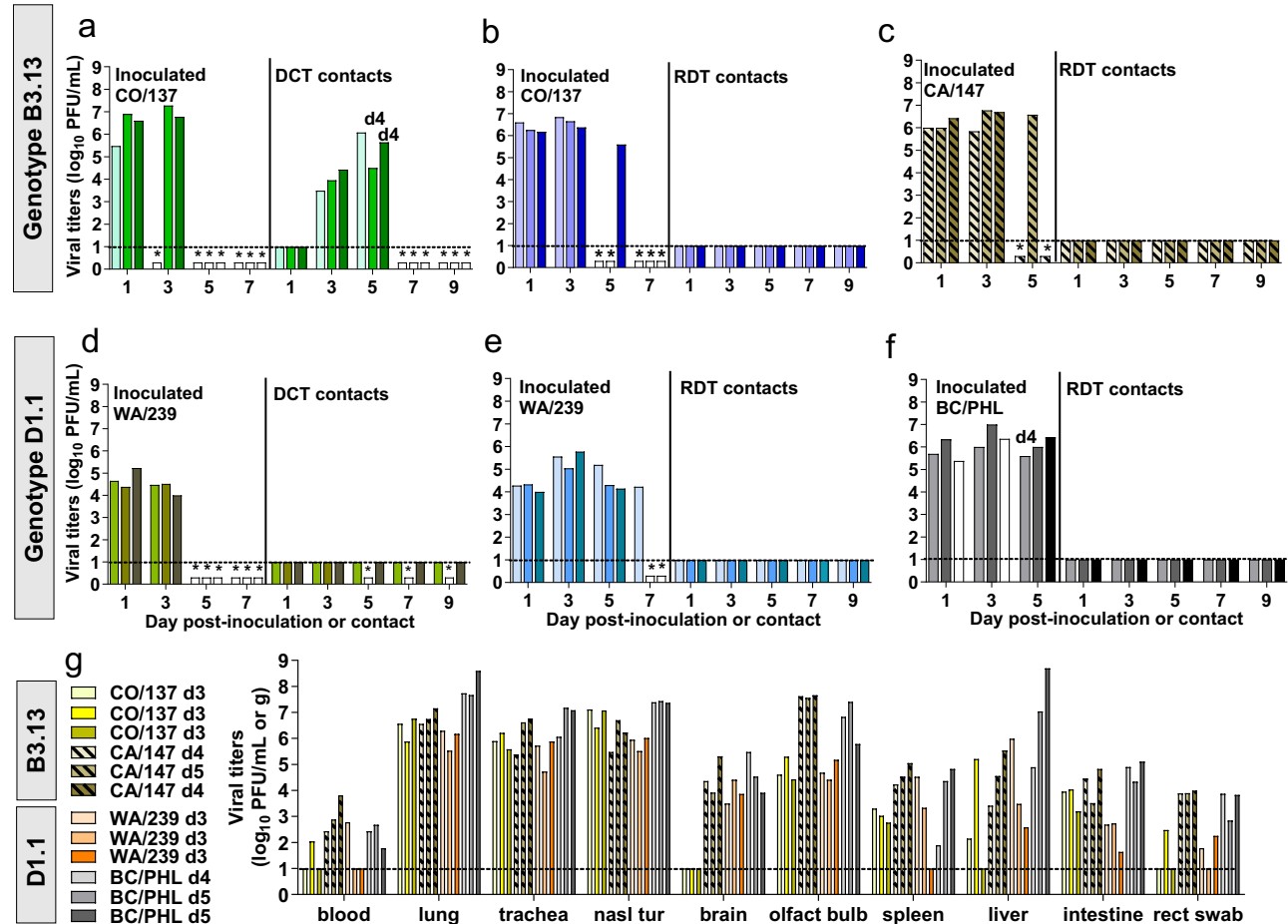

**Fig. 1 | Transmission and pathogenesis of genotype B3.13 and D1.1 A(H5N1) viruses in ferrets.** Ferrets ($n$ = 3–9) were inoculated with 6 log10 PFU of (**a**, **b**) A/Colorado/137/2024 (CO/137), (**c**) A/California/147/2024 (CA/147), (**d**, **e**) A/Washington/239/2024 (WA/239), (**f**) and A/British Columbia/PHL-2032/2024 (BC/PHL) viruses. Transmission was evaluated using the direct contact transmission model (DCT; 1 inoculated and 1 naïve ferret co-housed in one cage; transmission of CA/147 and BC/PHL were not tested), and the respiratory droplet transmission model (RDT; 1 inoculated and 1 naïve ferret housed in adjacent cages). Contact was established 24 h after inoculation of the donor ferrets. In graphs (**a**–**f**), bars on the left represent inoculated ferrets ($n$ = 3 per group), and the bars on the right represent contact ferrets ($n$ = 3 per group). Bars represent titers in nasal washes for individual animals collected every other day post-inoculation or contact, or on the day of euthanasia (day 4 indicated on top of the bar). White bars with asterisks denote animals that succumbed to infection prior to sample collection. **g** Dissemination of the virus to tissues was evaluated on day 3 post-inoculation for CO/137 ($n$ = 3; yellow bars) and WA/239- ($n$ = 3; orange bars) infected ferrets, or on the day the humane endpoint was reached for CA/147- and BC/PHL-inoculated ferrets (patterned and grey bars, respectively; the same ferrets as the inoculated ferrets shown in graphs **c**, **f**). Viral titers are reported as log10 PFU/mL [blood, nasal turbinates (nasl tur), rectal swabs (rect swab)], or log10 PFU/gram (lung, trachea, brain, olfactory bulb (olfact bulb), spleen, liver, intestine] of tissue. The limit of detection is 10 PFU/mL or g (dashed line). Each bar represents an individual ferret. The graphs were generated using GraphPad Prism and the source data has been deposited on GitHub[74].

disease and necessitated humane euthanasia between days 2–7 post inoculation (p.i.). To assess systemic spread in tissues, three separate ferrets per group were inoculated with a representative strain of each genotype, CO/137 (B3.13) and WA/239 (D1.1), and euthanized on day 3

p.i. (Fig. 1g). Although no additional animals were included for tissue distribution analysis on day 3 p.i., for CA/147 (B3.13) and BC/PHL (D1.1), tissues were collected from the inoculated ferrets used in the transmission experiments (shown in Fig.1c, f) at the time they reached

humane endpoint criteria (days 4–5 p.i.). Regardless of virus strain or time point p.i., high viral titers were consistently detected throughout the respiratory tract. Evidence of systemic spread in all animals was demonstrated by the presence of virus in other tissues including the olfactory bulb, intestine, spleen, and liver (Fig. 1g), which was consistent with previously studied A(H5N1) viruses that were lethal in ferrets[23].

To determine whether variants harboring mammalian adaptation markers emerged following infection in ferrets, we performed next-generation deep sequencing analyses of viruses shed in nasal washes and tissues from ferrets inoculated with representative strains of each genotype (B3.13 CO/137 and D1.1 WA/239). No genetic changes at known mammalian adaptation markers were detected in samples collected from inoculated animals, and the identified variants were not consistently enriched across groups (BioProject ID: PRJNA1338708). These findings indicate a lack of strong selective pressure against these viruses, with previous reports showing no apparent trends of mammalian adaptation in ferrets infected with other clade 2.3.4.4b A(H5N1) viruses[10,13,14].

## Transmission of genotype B3.13 and D1.1 A(H5N1) influenza viruses in the ferret model

Ferret transmissibility studies provide critical experimental evidence on the potential for human-to-human spread of novel influenza viruses, directly informing pandemic risk assessment rubrics[24,25]. In the direct contact transmission (DCT) model, an inoculated ferret is co-housed with a naïve ferret, allowing transmission via multiple routes: direct contact, indirect contact through fomites, and inhalation of airborne virus[26]. Previously evaluated genotype B3.13 viruses displayed capacity for transmission in the DCT model[12,14,27]. To determine whether other clade 2.3.4.4b viruses retain the capacity for transmission in this setting, we tested a representative strain of each genotype CO/137 (B3.13) and WA/239 (D1.1). Similar to what was observed in previously tested genotype B3.13 strains, CO/137 virus efficiently transmitted between each of the three cohoused ferret pairs as evidenced by the detection of virus in nasal wash samples (mean maximum of 5.7 $\log_{10}$ PFU/mL) and rectal swabs (mean maximum of 2.5 $\log_{10}$ PFU/mL) of contact animals (Fig. 1a). All ferrets that became infected with CO/137 virus developed fatal disease with clinical signs including fever, weight loss, lethargy, nasal discharge, and diarrhea. The virus was also detected in blood, lungs, and nasal turbinates (1.9–6.5 $\log_{10}$ PFU/mL or g) in all contact animals. In contrast, direct contact transmission was less apparent with the D1.1 isolate, WA/239. Although virus was not detected in any of the contact nasal wash samples, a rectal swab sample from one animal collected on day 3 post-contact (p.c.) had a titer of 3.4 $\log_{10}$ PFU/mL. This animal displayed weight loss, nasal discharge, diarrhea, and lethargy and reached endpoint criteria on day 3 p.c. (Fig. 1d, Supplementary Fig. 2d). Blood, lung, and trachea samples collected at the time of euthanasia had detectable virus (1.3–2.0 $\log_{10}$ PFU/mL or g), while the nasal turbinate sample did not. The remaining two ferrets did not display signs of infection and did not have detectable virus in nasal wash and rectal swab samples: however, one seroconverted to WA/239 (HAI 160 on day 21 p.c.). These data show a higher transmission capacity of the B3.13 virus as compared to the D1.1 virus in a direct contact setting.

Next, we used the respiratory droplet transmission model (RDT) to determine if recently isolated A(H5N1) viruses (CO/137, CA/147, WA/239, and BC/PHL) were capable of airborne transmission (Fig. 1b, c, e, f). In this set-up, the inoculated and contact ferrets are housed in adjacent cages that permit air exchange but prevent physical contact[26]. Three ferret pairs were used for each virus group. None of the four tested viruses were capable of airborne transmission as evidenced by the lack of virus detection in nasal wash and rectal swab samples, and the lack of seroconversion to the challenge virus. Notably, earlier B3.13 viruses were capable of limited airborne transmission, suggesting both intra- and inter-genotypic variability in transmission phenotypes[12–14].

## Comparative evaluation of airborne virus shedding using BC251 and SPOT samplers

To better understand the relationship between airborne virus shedding and transmission in the ferret RDT model, we used a panel of influenza viruses representing a range of airborne transmissibility profiles (Table 1). The viruses were categorized into three groups based on their known transmission frequencies: non-transmissible, low-transmissible, and high-transmissible. In addition to testing clade 2.3.4.4b A(H5N1) strains (B3.13 genotype: CO/137, CA/147 and D1.1 genotype: WA/239, BC/PHL), we included a non-transmissible predecessor strain, clade 1 A(H5N1) A/Vietnam/1203/2004 (VN/1203) strain, for comparison. Two previously characterized low-transmissible strains were also included: clade 2.3.4.4b A(H5N1) (MI/90; genotype B3.13), which transmitted between 50% of ferret pairs[12], and A(H7N9) A/Anhui/1/2013 (Anhui/1), which transmitted in 33% of pairs[28]. Finally, two high-transmissible viruses were tested: human seasonal A(H1N1)pdm09 A/Nebraska/14/2019 (NE/14) and A(H9N2) A/Anhui-Lujiang/39/2018 (AL/39), which transmitted between 100% and 83% of tested ferret pairs, respectively[29,30]. Since different air sampling devices have varying collection efficiencies and preserve viral particle infectivity to differing degrees[19], we employed two aerosol sampling devices for comparison. The NIOSH BC251 two-stage cyclone sampler collected air at 3.5 L/min and size-fractionated particles into a 15 mL tube (> 4 μm), a 1.5 mL tube (1–4 μm), and a 37 mm PTFE filter (< 1 μm). Because most viral RNA and infectious particles were recovered in the >4 μm fraction, consistent with a previous study employing diverse influenza virus strains[18], we report totals combined across all fractions. The water condensation capture-based SPOT sampler collected particles into liquid media at 1.5 L/min. Air was collected sequentially from inoculated ferrets (number of replicates and collection times are shown in Supplementary Table 1) and the data were normalized to sampled air volume as described in the Methods section to permit direct comparison between samplers and to account for sampling time and flow rate differences. Each of the aerosol sampling devices was capable of capturing both genomic RNA and infectious virus; however, differences in the detection frequencies and viral titers were observed, indicating variability in sampler-specific collection efficiencies as well as virus strain-specific variation in the amount of airborne virus shed by infected animals. In all air samples with detectable virus, genomic RNA copy titers consistently exceeded infectious virus titers (Fig. 2, Supplementary Fig. 3). Out of 184 samples generated using the BC251 sampler throughout all experiments, infectious virus and viral RNA were detected in 68% and 93% of samples, respectively. In comparison, out of 97 samples collected using the SPOT sampler, infectious virus and viral RNA were detected in 52% and 77% of samples, respectively. Fisher's Exact Test analysis showed that the odds of virus detection with the BC251 sampler were significantly higher than with the SPOT sampler (infectious virus: $p \leq 0.006$, genomic RNA: $p \leq 0.0002$). Specifically, the BC251 sampler was 2 times more likely to detect infectious virus and 4.2 times more likely to detect viral RNA than the SPOT sampler (Supplementary Table 2). Furthermore, the BC251 sampler collected genomic RNA and infectious virus from transmissible viruses at significantly higher levels compared to the SPOT sampler, supportive of better collection efficiency of the BC251 sampler (Fig. 3a, b; high-transmissible: $p \leq 0.0005$, low transmissible: $p \leq 0.0008$, non-transmissible: $p \geq 0.07$). It is important to note that without normalization to the sampled air volume, the statistical differences between the samplers would have been even greater, as the SPOT sampler drew about 2.33 times less air per minute compared to the BC251 sampler. To assess the differences in sampler-specific retention of virus viability during the collection, we compared the ratios of genomic RNA to PFU titers in samples where both measurements were detected. Lower mean RNA/PFU ratios were observed in samples collected by the SPOT sampler relative to the BC251 sampler, indicative of higher retention of virus viability when

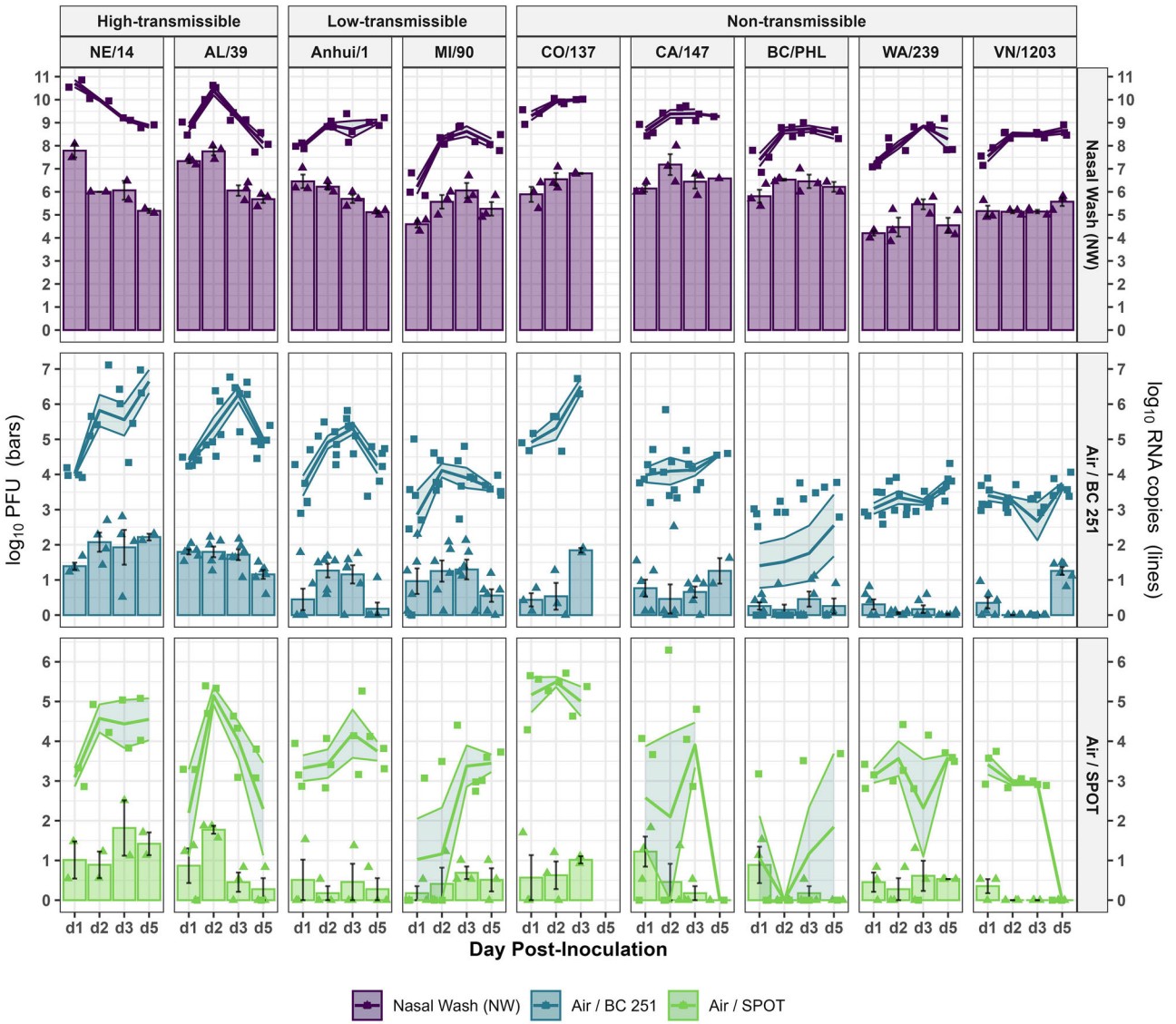

**Fig. 2 | Detection of airborne influenza virus shed by inoculated ferrets.** Ferrets were inoculated with 6 log$_{10}$ PFU of A(H1N1)pdm09 A/Nebraska/14/2019 (NE/14) ($n$ = 2), A(H9N2) A/Anhui-Lujiang/39/2018 (AL/39) ($n$ = 3), A(H7N9) A/Anhui/1/2013 (Anhui/1) ($n$ = 3), A(H5N1) A/Michigan/90/2024 (MI/90) ($n$ = 3), A(H5N1) A/Colorado/137/2024 (CO/137) ($n$ = 3), A(H5N1) A/California/147/2024 (CA/147) ($n$ = 3; donors in RDT experiment), A(H5N1) A/British Columbia/PHL-2032/2024 (BC/PHL) ($n$ = 3; donors in RDT experiment), A(H5N1) A/Washington/239/2024 (WA/239) ($n$ = 3; donors in RDT experiment), and A(H5N1) A/Vietnam/1203/2004 (VN/1203) $n$ = 3). Virus titers in each sample were evaluated using both standard plaque assay to determine infectious virus load (individual points-triangles, mean-bars) and real time qRT-PCR to determine genomic RNA copy load (individual points-squares,

means-lines). The air data are presented as log$_{10}$ PFU or RNA copies per 105 liters of air to standardize between both samplers, and the nasal wash data are presented as PFU/mL and RNA copies/mL. Virus titers in nasal wash samples collected from individual inoculated ferrets are shown in purple, virus titers in air samples collected by BC251 sampler are shown in teal, and virus titers in air collected by a SPOT sampler are shown in green. AL/39 d5 data timepoint indicates data collected on day 4. Shaded areas and error bars represent the standard error of the mean, calculated from all replicates, and are shown for descriptive purposes only to help visualize data variability for each sample type and are not used to draw statistical conclusions. The graphs were generated using R and the source data and code has been deposited on GitHub[74].

using the SPOT sampler. However, this difference reached statistical significance only in samples from ferrets inoculated with the high-transmissible virus group ($p \leq 0.02$) (Fig. 3c). Overall, both samplers demonstrated high utility for influenza virus aerosol collection, each offering distinct advantages, suggesting that their parallel use can provide a more comprehensive understanding of airborne virus dynamics across a wide range of viruses.

### Correlation between influenza virus loads in ferret nasal washes and air samples

In our previous study, we detected significantly higher levels of genomic RNA in nasal washes and air samples collected from ferrets inoculated with transmissible viruses compared to those inoculated

with non-transmissible viruses[18]. However, since the presence of influenza genomic RNA does not necessarily indicate the presence of viable, infectious virus, in this study, we performed side-by-side comparisons of both viral RNA and infectious virus shed by inoculated ferrets. To measure the correlation between virus exhaled into the air and virus load in nasal washes, the area under the curve (AUC) was calculated for samples collected over days 1, 2, and 3 p.i. (day 5 was omitted due to an incomplete dataset). This approach was selected to allow comparisons between viruses displaying differing peak shedding times. A strong positive correlation using Pearson's method was observed between AUC values, with R = 0.7 for infectious virus and R = 0.82 for genomic RNA (Fig. 4a. b), suggesting that higher viral loads in nasal wash specimens during the acute phase of infection are

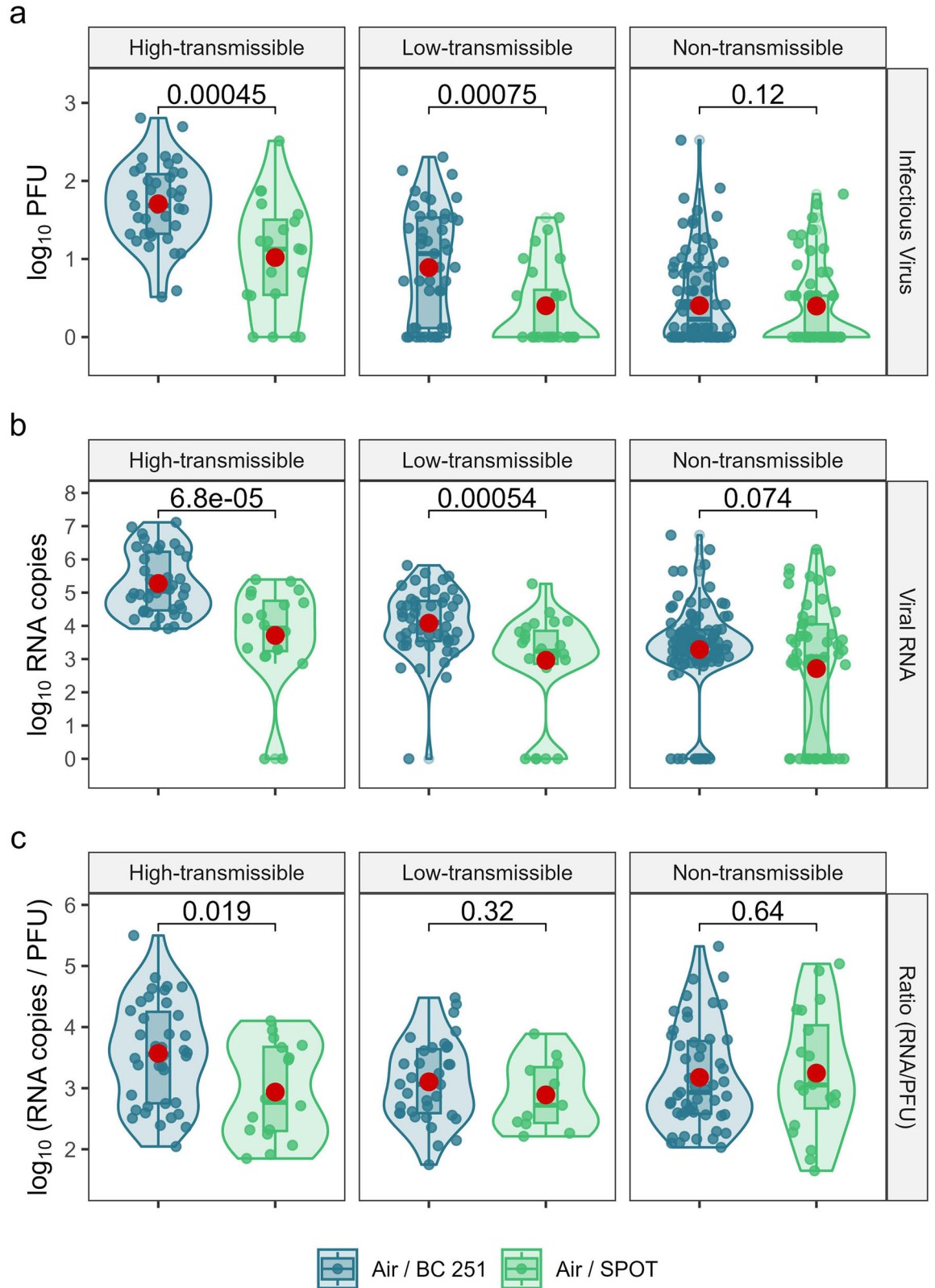

associated with increased virus exhaled into the air (collected by the BC251 samplers). Similar trends were observed for the SPOT sampler, although the correlations were more modest (R = 0.61 for infectious virus and R = 0.64 for genomic RNA; Supplementary Fig. 4a, b). To better understand the nature of this correlation, it was important to determine whether it was driven by viral loads on specific days or by cumulative viral shedding over multiple days, as represented by the AUC. When viral loads from individual days were compared between nasal wash and air samples collected by BC251 or SPOT samplers using Pearson's method, positive correlations were still observed, albeit reduced (R = 0.5–0.21 for infectious virus and R = 0.57–0.33 for genomic RNA; Supplementary Fig. 5). When linear mixed-effects model was utilized with a random intercept for individual ferret, nasal wash to airborne virus collected by BC251 slopes were near zero on day 1 and

**Fig. 3 | Comparison of sampling efficiencies between BC251 and SPOT samplers.** Viruses were grouped based on their airborne transmissibility profiles into three categories: high-transmissible (A/Nebraska/14/2019 and A/Anhui-Lujiang/39/2018; 100-83% transmission frequency), low-transmissible (A/Michigan/90/2024 and A/Anhui/1/2013; 50-33% transmission frequency), and non-transmissible (A/Colorado/137/2024, A/California/147/2024, A/Washington/239/2024, A/British Columbia/PHL-2032/2024, and A/Vietnam/1203/2004; 0% transmission frequency). Pairwise comparisons of (**a**) infectious virus (PFU) and (**b**) genomic RNA copies collected per 105 L of air using BC251 (teal) and SPOT (green) samplers are shown for each group ($n \geq 20$ biological replicates). **c** Ratios of RNA copy number to PFU were calculated for samples where both measurements were detected ($n \geq 11$ biological replicates). Statistical analyses were performed using the two-sided Wilcoxon Rank Sum test, with $p$-values shown above boxplots. Boxplots represent interquartile range (25th–75th percentiles); median (center line); mean (red dot); the whiskers represent the minimum/maximum values within 1.5 x the interquartile range; points beyond the whiskers represent outliers. The graphs were generated using R and the source data and code has been deposited on GitHub[74].

day 2 and positive on day 3 (infectious virus: 0.369, 95% CI 0.047 to 0.692 and genomic RNA: 0.857, 95% CI 0.156–1.558). Tukey-adjusted pairwise comparisons of slopes showed trends toward stronger day 3 associations versus day 1 (infectious virus: 0.389, $p = 0.074$; genomic RNA: 0.805, $p = 0.064$), with other contrasts not significant. These findings indicate that while viral loads in nasal washes on individual days show some correlation with airborne virus levels, the cumulative viral shedding captured by the AUC provides a more robust and reliable measure when comparing viruses with different shedding kinetics. When comparing across transmissibility groups, AUC means for nasal wash and air samples from animals infected with highly transmissible viruses were significantly higher compared to those from animals infected with non-transmissible viruses. Low-transmissible viruses generally displayed AUCs that were not statistically different from those of either high- or non-transmissible viruses, indicating a more variable shedding phenotype in this intermediate group (Fig. 4, Supplementary Table 3). This trend was consistent across both infectious virus and genomic RNA datasets. Taken together, these findings strengthen the central conclusion that transmissibility is linked to the magnitude of virus shed in nasal washes and the surrounding air.

## Probability of airborne transmission of A(H5N1) influenza between ferrets based on virus shedding measurements

To better evaluate the pandemic risk associated with A(H5N1) influenza viruses, we classified the nine tested viruses into two groups, transmissible and non-transmissible, based on their observed transmission outcomes (with viruses associated with airborne transmission between at least one ferret pair classified as transmissible[15]). We used AUC values, calculated from infectious virus titers and viral RNA copy numbers in BC251 air samples collected on days 1, 2, and 3 post-inoculation to model the probability of airborne transmission using logistic regression with transmissibility as a binary parameter (Fig. 5a, b). We then used the fitted model to calculate the probability of airborne transmission for each virus (Fig. 5c). The non-transmissible D1.1 A(H5N1) viruses, WA/239 and BC/PHL, as well as the clade 1 A(H5N1) VN/1203 virus, had predicted airborne transmission probabilities of $\leq 16\%$ (Fig. 5c). In contrast, highly transmissible NE/14 and AL/39 viruses showed a high predicted transmission potential of over 80%. Interestingly, viruses that did not transmit between ferret pairs, such as B3.13 CA/147 and CO/137, showed a wide range of predicted transmission probabilities (19–95%) similar to what was observed for MI/90 and Anhui/1 which were previously reported to transmit between 33% and 50% of ferret pairs, highlighting the variability in airborne viral shedding among ferrets inoculated with viruses that are not capable of efficient airborne transmission. These findings show that although both D1.1 (WA/239, BC/PHL) and B3.13 (CA/147, CO/137) viruses did not transmit in the ferret RDT model in this study, B3.13 strains were shed into the air at higher levels, and therefore had higher overall calculated probabilities of airborne transmission potential compared to D1.1 viruses, supporting the utility of assessing airborne virus shedding in the context of pandemic risk assessment.

## Discussion

This study offers important insights into the pathogenesis and transmission potential of recent HPAI A(H5N1) from the B3.13 and D1.1 genotypes, which were linked to outbreaks in poultry and dairy cattle in 2024[5]. Using ferrets, a well-established model for influenza virus infection, we observed a uniformly severe disease phenotype among all tested viruses (B3.13: CO/137, CA/147, and D1.1: WA/239, BC/PHL), regardless of genotype or human exposure source. The disease progressed rapidly in ferrets, marked by fever, weight loss, lethargy, respiratory symptoms, and frequent diarrhea. Systemic spread was demonstrated by the detection of virus in blood, liver, olfactory bulbs, gastrointestinal tissues, and other organs, consistent with previous findings for B3.13 viruses that were lethal in ferrets[13,21,31]. These results highlight that both B3.13 and D1.1 viruses possess the capacity to cause systemic disease in susceptible mammals. However, the findings do not preclude the existence of less virulent strains within these genotypes, as non-lethal strains, such as B3.13 genotype MI/90, have been previously reported[12]. This variability underscores the need to identify specific viral determinants of lethality, particularly given the high genetic similarity among viruses within genotypes.

The ferret transmission data further distinguish B3.13 from D1.1 viruses. CO/137 (B3.13) transmitted efficiently in the presence of direct contact, as evidenced by virus detection in nasal washes and rectal swabs, with all contact animals ultimately developing fatal disease. This is in agreement with previous observations of transmissibility in B3.13 strains (TX/37 and MI/90)[12,14], indicating this phenotype is likely shared among genetically similar B3.13 viruses. In contrast, WA/239 virus (D1.1) exhibited limited transmission between cohoused ferrets. Among the three contact ferrets, virus was detected in rectal swab, blood, lung, and trachea samples from one ferret. Since the virus was not detected in nasal washes or nasal turbinates, this suggests potentially less efficient replication in the upper respiratory tract for D1.1 viruses. Another contact ferret seroconverted to WA/239 in the absence of virus detection in nasal wash or rectal swab samples. Seroconversion in the absence of virus detection has been documented for other H5 subtype viruses[32,33]. Overall, inefficient transmission in a direct contact setting has occasionally been reported with earlier clade A(H5N1) viruses[16,26,34] and with other more recent clade 2.3.4.4b viruses[10,32] indicating that, although relatively rare, transmission in the presence of direct contact can occur with certain A(H5N1) strains.

Despite clear evidence of transmission using the more permissive direct contact transmission model, none of the tested viruses, including B3.13 strains, transmitted via air. This absence of airborne transmission of the B3.13 viruses contrasts with previous reports showing limited transmission in the ferret model, but it is consistent with the historical phenotype of H5 subtype viruses, which typically require extensive adaptation, such as changes in HA receptor binding, polymerase activity, or pH stability, to spread efficiently via the air in mammals[35,36]. We previously showed that transmissible B3.13 strains were detected in the air at higher frequencies compared to a non-transmissible B3.2 genotype A(H5N1) strain (A/Chile/25945/2023)[12,14]. To further investigate whether the lack of airborne transmission observed in this study was due to the lack of infectious virus in the air, we used BC251 and SPOT samplers to quantify levels of infectious virus shed into the air and to compare these levels with those observed for other transmissible viruses. Air samples from ferrets infected with highly transmissible viruses showed significantly higher infectious virus and genomic RNA levels compared to non-transmissible strains,

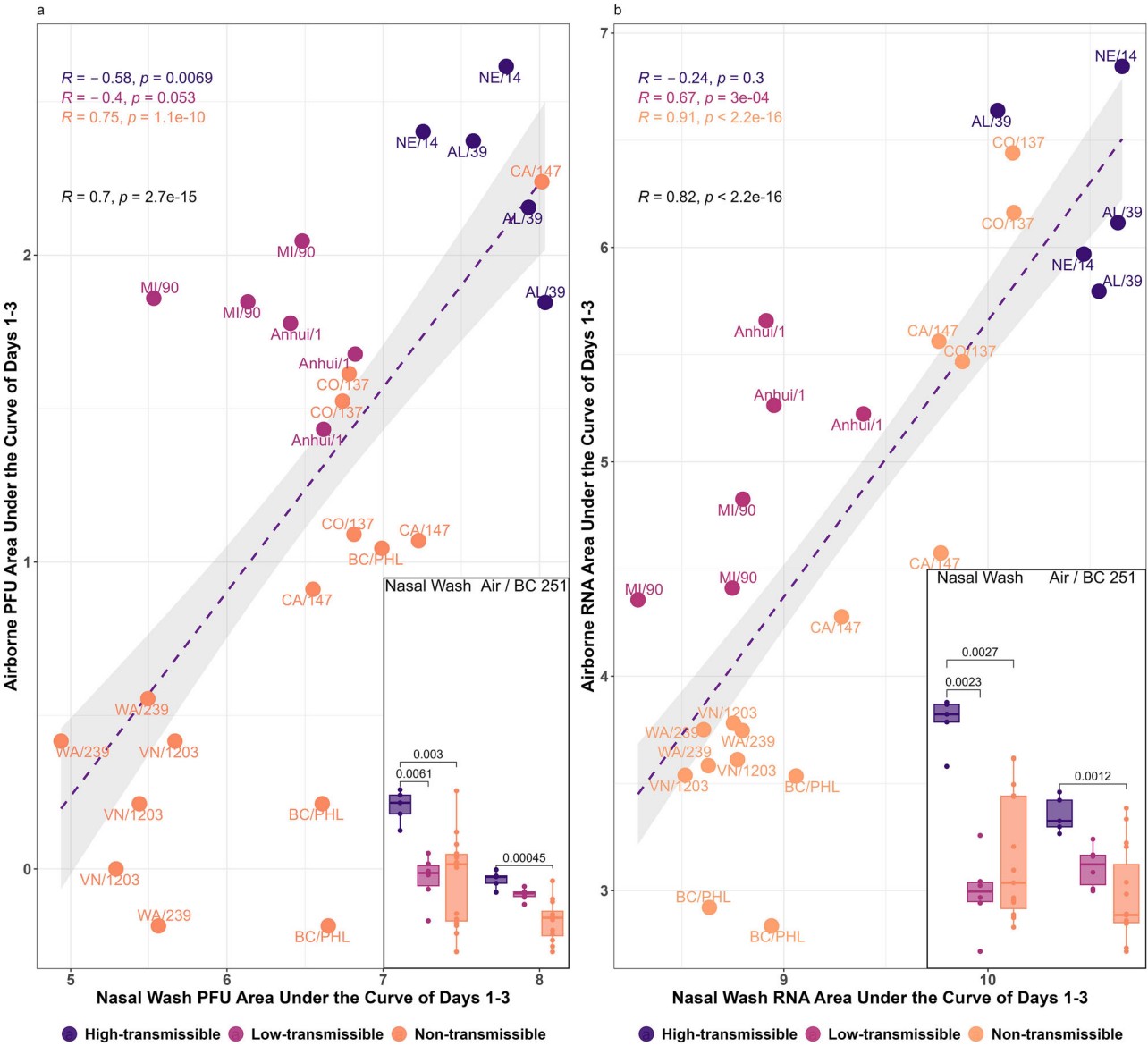

**Fig. 4 | Correlation between viral loads in nasal washes and air samples from inoculated ferrets using BC251 sampler.** Area under the curve (AUC) values were calculated using infectious virus (PFU) (**a**) and viral RNA copy (**b**) titers from nasal wash and air samples collected on days 1, 2, and 3 post-inoculation with 6 log$_{10}$ PFU of A(H1N1)pdm09 A/Nebraska/14/2019 (NE/14) ($n = 2$), A(H9N2) A/Anhui-Lujiang/ 39/2018 (AL/39) ($n = 3$), A(H7N9) A/Anhui/1/2013 (Anhui/1) ($n = 3$), A(H5N1) A/ Michigan/90/2024 (MI/90) ($n = 3$), A(H5N1) A/Colorado/137/2024 (CO/137) ($n = 3$), A(H5N1) A/California/147/2024 (CA/147) ($n = 3$), A(H5N1) A/British Columbia/PHL-2032/2024 (BC/PHL) ($n = 3$), A(H5N1) A/Washington/239/2024 (WA/239) ($n = 3$), and A(H5N1) A/Vietnam/1203/2004 (VN/1203) ($n = 3$). Viruses were grouped based on their known airborne transmissibility profiles in the ferret model into three categories: high-transmissible (NE/14 and AL/39; 100-83% transmission frequency;

purple circles and bars; $n = 5$), low-transmissible (MI/90 and Anhui/1; 50-33% transmission frequency; magenta circles and bars; $n = 6$), and non-transmissible (CO/137, CA/147, BC/PHL, WA/239, VN/1203; 0% transmission frequency; orange circles and bars; $n = 15$). Pearson's correlations between AUC values are represented by the dashed line, with 95% confidence intervals shown as shading. Pairwise comparisons were conducted using the Kruskal-Wallis test with Dunn's multiple comparisons post-hoc analysis; high-transmissible ($n = 5$), low-transmissible ($n = 6$), and non-transmissible ($n = 15$). Boxplots represent interquartile range (25th–75th percentiles); median (center line); the whiskers represent the minimum/maximum values within 1.5 x the interquartile range; points beyond the whiskers represent outliers. The graphs were generated using R and the source data and code has been deposited on GitHub[74].

which was consistent with previous studies employing different air collection protocols and virus strains[11,16,20]. This supports the conclusion that efficient upper respiratory tract replication and robust release of infectious virus into the air are critical for efficient airborne dissemination[37–39]. In addition, this finding aligns with previous reports suggesting that airborne viral load measurements can help predict virus transmissibility and support the early identification of viruses with pandemic potential[11,18]. Notably, overlapping shedding profiles were sometimes observed between transmissibility groups, suggesting a spectrum of phenotypes influenced by additional viral or host

factors. Although CA/147 and CO/137 viruses (genotype B3.13) were not transmissible in the ferret RDT model, unlike earlier strains of the same genotype, MI/90 and TX/37, these viruses were shed into the air at significantly higher levels than the D1.1 viruses, as evidenced by significantly higher AUC means (PFU $p = 0.00813$; RNA $p = 0.00216$). Also, B3.13 viruses exhibited a broader range of predicted transmission probabilities, similar to strains previously shown to transmit inefficiently though the air. Overall, the data shows that the lack of observed transmission was not because of the absence of airborne virus in the air. One potential reason may be decreased infectivity of these strains,

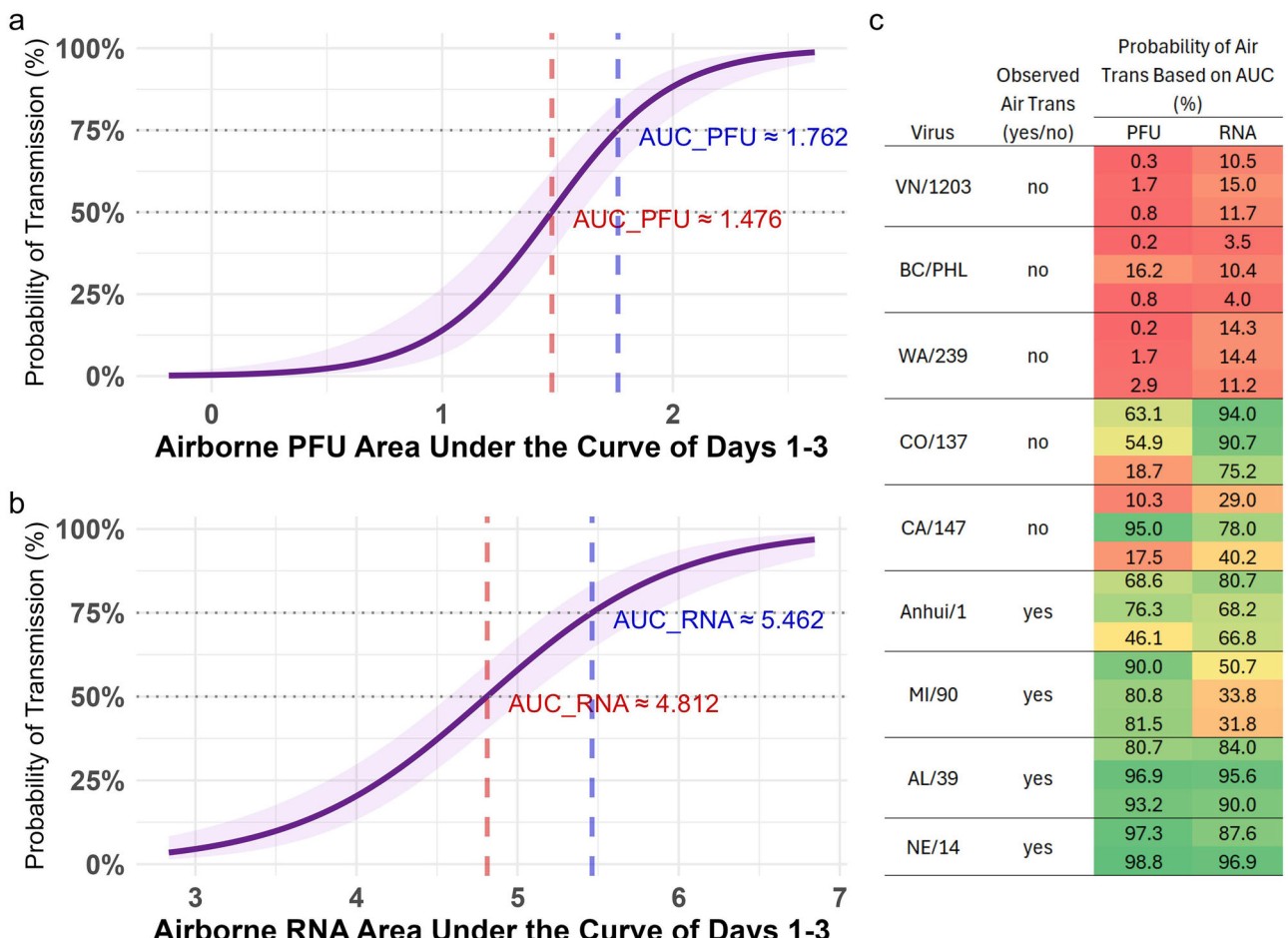

**Fig. 5 | Calculated probabilities of airborne transmission of diverse influenza viruses in the ferret model.** Area under the curve (AUC) values based on infectious virus titers (PFU) (**a**) and viral RNA copy numbers (**b**) from BC251 air samples collected on days 1, 2, and 3 post-inoculation were used, along with transmission outcomes (yes/no), to generate probability curves using a logistic regression model. Solid purple lines represent the fitted regression curves, with purple shaded regions indicating 95% confidence intervals. Dashed lines denote AUC thresholds corresponding to 50% (red) and 75% (blue) predicted probabilities of airborne transmission. **c** Calculated percent probability of a virus possessing airborne transmission potential based on measured airborne virus levels, with probabilities ranging from low (red) to high (green). The graphs were generated using R and the source data and code has been deposited on GitHub[74].

necessitating a higher inoculation dose to initiate productive infection[40]. Investigation of the relationship between airborne influenza shedding and the airborne infectious dose in ferrets may further shed light on transmissibility[20].

Recent studies demonstrated that specific mutations, leading to binding of human-like receptors, increased HA acid stability, and enhanced replication in mammalian cells, can play a role in airborne A(H5N1) virus shedding. This highlights the combined effect of multiple factors on transmissibility[11]. Sequence analysis of the HA gene of the A(H5N1) viruses studied here revealed that key residues at positions 190, 226, and 228 (H3 numbering) retain avian-type molecular signatures, consistent with strong α2,3-linked sialic acid binding preference demonstrated in previous B3.13 A(H5N1) virus studies[14,41] (Supplementary Table 4). A limited number of mammalian adaptation markers were detected in PB2 and PA, known to enhance replication in mammalian hosts[42–44]. Notably, the PB2 G362E substitution was detected in the WA/239 and BC/PHL viruses, as well as in the previously studied TX/37 and VN/1203 strains. The PB2 E627K substitution was identified in the PB2 gene of BC/PHL, TX/37, and VN/1203. Additionally, the PB2 M631L and PA K497R substitutions were observed in the CO/137 and CA/147 virus sequences and have also been reported in the previously studied MI/90 strain. A pertinent question is whether variants that could drive transmission and/or pathogenesis emerge in ferrets during infection. Mutational frequencies observed in the ferret

model may provide insights into the transmission potential in humans[45]. Deep sequencing analyses of viruses shed in nasal washes and tissues from ferrets inoculated with the CO/137 (B3.13) and WA/239 (D1.1) strains revealed no genetic changes at known mammalian adaptation marker locations. These findings support a lack of strong selective pressure against these viruses, aligning with previous reports indicating no apparent trends toward mammalian adaptation in ferrets infected with other clade 2.3.4.4b A(H5N1) viruses[10,13,14]. Studies employing techniques optimized for low-titer specimens may further asses mutational frequencies in air samples collected from ferrets inoculated with diverse influenza virus strains. Collectively, while the identification of genetic markers associated with mammalian adaptation offers valuable insights, they do not fully account for the observed differences in airborne virus shedding and transmission capabilities among the clade 2.3.4.b A(H5N1) viruses tested here and previously[12–14]. This underscores the need for further molecular investigations, particularly in mammalian model systems, to identify additional genetic determinants of airborne transmissibility. In addition, continued genetic surveillance is essential, as even a single HA mutation can switch receptor specificity to human-type sialic acids and significantly influence mammalian adaptation of viruses[46].

The choice of aerosol sampling devices is critical for both effectiveness and experimental feasibility, including the ability to efficiently quantify infectious virus. Earlier work suggested that the samplers

collecting aerosolized infectious virus particles into liquid media outperform samplers collecting particles onto dry surfaces[47]. Through direct comparison of two air sampling platforms, we found that each offers distinct advantages. The SPOT sampler, which uses water-based condensation capture, exhibited better retention of infectious virus, making it ideal for studies prioritizing virus viability or long-duration sampling[11]. Meanwhile, the BC251 sampler, which collects particles onto dry surfaces, showed lower viability retention but achieved greater overall detection sensitivity, especially at low airborne virus concentrations. Its simplicity, scalability, and suitability for high-throughput applications make it operationally advantageous. The complementary strengths of these platforms highlight their combined potential to enhance our understanding of airborne virus dynamics across diverse influenza viruses. Aggregating such data can support the development of mathematical models aimed at identifying strains with increased transmission potential and ultimately reducing the number of animals needed for risk assessments, in compliance with the 3Rs principle[48,49].

While this study advances risk assessment of the B3.13 and D1.1 genotype A(H5N1) viruses in ferrets through comparative evaluation of airborne virus shedding patterns, further studies using alternative experimental designs are warranted. We utilized a high intranasal inoculation dose of $10^6$ PFU/mL in serologically naïve animals allowing for direct comparison with a substantial body of data for diverse viruses generated through similar protocols[15]. Our focus was on virus shedding primarily during days 1 to 3 post-inoculation, a timeframe prior to reaching humane endpoint criteria, which generally coincided with peak nasal wash titer and fever onset (Supplementary Table 5). This emphasis on the early days post-infection is also supported by prior ferret studies, which indicate that transmission is more likely to occur before the onset of clinical signs in inoculated animals, and when contact with recipient animals is established early post inoculation[50–52]. However, it is important to note that lower, more physiologically relevant inoculation doses and routes (e.g., aerosol inhalation route) may lead to delayed and/or reduced shedding peaks and influence transmission likelihood[53]. The infectious dose required for productive infection may also vary by strain, introducing additional variability[20,40]. Our comparative analyses using a sizable and diverse panel of viruses to identify shedding trends in both nasal washes and air at a set dose provide a solid framework for follow-up studies that could explore the feasibility of testing airborne virus shedding following lower inoculation doses and other pertinent variables.

Collectively, the findings in this study underscore the ongoing threat posed by B3.13 genotype viruses, which consistently transmit in the presence of direct contact in ferrets and shed higher levels of airborne virus compared to other non-airborne-transmissible strains. While D1.1 viruses show lower transmission potential, their widespread circulation in poultry and periodic spillover into dairy cattle highlight the need for continued surveillance. In addition, although current A(H5N1) viruses pose limited risk for human-to-human spread, repeated mammalian exposure could enable future adaptation. Lastly, our use of two complementary aerosol collection methods in ferrets provides new insights into how shedding of infectious virus contributes to transmission. Incorporating aerobiological analyses into risk assessments will improve our ability to detect emerging strains with greater airborne virus shedding and transmission potential.

## Methods

### Biosafety

The work was conducted to inform public health risk assessments. All animal experiments, including those involving HPAI viruses, were performed under enhanced biosafety level 3 (BSL-3E) containment in a facility under continuous 24/7 monitoring. Facility access was restricted to trained personnel, authorized to enter the laboratory suite as regulated by the Federal Select Agent Program. Enhancements of the BSL-3E suite included HEPA filtration on supply and double HEPA filtration on the exhaust air. All research activities adhered to safety and security protocols established by the U.S. Department of Agriculture and the Select Agent Program, in accordance with the Biosafety in Microbiological and Biomedical Laboratories guidelines[54]. The work was conducted under strict adherence to administrative controls and the use of personal protective equipment. Biosafety cabinets and secondary containment devices around animal caging were utilized for all work with infectious material. Wild-type viruses isolated and propagated from clinical samples were used for ferret inoculations, and all ferret samples collected in this study were destroyed upon completion of this work.

### Virus stocks

Virus stocks of A/Colorado/137/2024 (CO/137), A/Washington/239/2024 (WA/239), A/British Columbia/PHL-2032/2024 (BC/PHL-2032), A/Michigan/90/2024 (MI/90), and A/Nebraska/14/2019 (NE/14) were propagated in MDCK cells at 37 °C for 48 h. A/California/147/2024 (CA/147), A/Vietnam/1203/2004 (VN/1203), A/Anhui/1/2013 (Anhui/1), and A/Anhui-Lujiang/39/2018 (AL/39) were grown in the allantoic cavities of 10-day-old embryonated chicken eggs incubated at 37 °C for 24–26 h; allantoic fluid from multiple eggs was pooled. Supernatants were clarified by centrifugation, aliquoted, and stored at −80 °C[55]. Viral titers were determined by standard plaque assay, and full-genome sequencing was performed to confirm subtype identity and exclude contamination with other influenza strains.

### Ferret experiments

All animal studies were conducted in accordance with protocols approved by the Centers for Disease Control and Prevention (CDC)'s Institutional Animal Care and Use Committee, at an AAALAC International-accredited facility. Fitch ferrets (Mustela putorius furo, Triple F Farms, Sayre, PA), aged 5–10 months, were used, and all animals were male, as sex-related variability has not been shown to affect data reproducibility across influenza risk assessment studies[56]. The animals were randomly assigned to experimental groups. Smaller animal groups were used for previously studied viruses, while larger groups were used for newly characterized viruses: NE/14 $n = 2$, AL/39 $n = 3$, Anhui/1 $n = 3$, MI/90 $n = 3$, VN/1203 $n = 3$, CA/147 $n = 6$, BC/PHL $n = 6$, WA/239 $n = 15$, and CO/137 $n = 15$. The sample sizes were consistent with previously established protocols and published work[57]. Using fewer ferrets per virus allowed us to conduct a greater total number of experiments and evaluate multiple viruses within the constraints of ethical animal use. By grouping viruses according to their transmissibility phenotypes, we maintained sufficient sample sizes within each group to ensure robust statistical analyses. Animal studies were not repeated. Prior to study initiation, all ferrets were confirmed seronegative for circulating influenza A and B viruses by hemagglutination inhibition (HAI) assay.

Ferrets were housed individually in stainless steel cages (56 × 42 × 42 cm) with soft pellet bedding, placed inside Duo-Flo Bioclean mobile units (Lab Products Inc., Seaford, DE) that provided 150–180 air changes per hour. Ferrets were anesthetized before transponder placement, virus inoculation, nasal and rectal sample collection, and euthanasia. Anesthesia was administered intramuscularly using a combination of ketamine (10–30 mg/kg) and xylazine (1-2 mg/kg). Prior to inoculation, a subcutaneous temperature transponder (IPTT-300, BMDS) was inserted into the dorsal space between the scapulae of each animal. Baseline temperature and weight measurements were obtained one day prior to inoculation. Two to nine ferrets per group were intranasally inoculated with 6 $\log_{10}$ PFU of virus diluted in 1 mL PBS [CO/137 ($n = 9$), CA/147 ($n = 3$), WA/239 ($n = 9$), BC/PHL ($n = 3$), VN/1203 ($n = 3$), MI/90 ($n = 3$), Anhui/1 ($n = 3$), AL/39 ($n = 3$), NE/14 ($n = 2$)]. To assess transmissibility of A(H5N1) viruses, uninfected (naïve) ferrets were introduced 24 hours post-inoculation at a 1:1 ratio.

In the DCT model, naïve ferrets were co-housed with inoculated animals (3 pairs). In the RDT model, the ferret pairs were housed in adjacent cages separated by perforated walls that allowed airflow but prevented physical contact (3 pairs)[26]. Transmission in ferrets was determined by the detection of infectious virus and/or seroconversion. Nasal washes and rectal swabs were collected every 1–2 days from all inoculated and contact animals and stored at −80 °C[58]. All ferrets were observed daily or twice-daily post-inoculation for clinical signs of infection. Animals exhibiting persistent severe illness (e.g., diarrhea, lethargy, labored breathing) were humanely euthanized under anesthesia via intracardiac administration of 1 mL/kg (390 mg pentobarbital sodium and 50 mg phenytoin sodium per mL). Blood samples were collected into BD Vacutainer Plastic Blood Collection Tubes with Lithium Heparin (Becton, Dickinson and Company) on the day of euthanasia to assess viremia stored at −80 °C. Tissues were harvested from the euthanized animals to evaluate viral replication and systemic spread of the B3.13 and D1.1 A(H5N1) viruses. Three separate ferrets per group were used to assess systemic dissemination on day 3 p.i. for WA/239 and CO/137, following a timeline consistent with prior studies[12,14,23,32]. To minimize the number of animals used in this study, the systemic dissemination of CA/147 and BC/PHL was evaluated using euthanized donors from the RDT experiment on the day the animals reached humane endpoint criteria (days 4–5). The tissues were stored at −80 °C and subsequently thawed, weighted, and homogenized using Precision Disposable Tissue Grinders (Covidien LLC). The homogenates were clarified by centrifugation at 4 °C, and infectious virus titers were determined by standard plaque assay using MDCK cells, along with nasal wash, rectal swab, and whole blood samples.

## Air sampling

Air samples were collected from ferrets inoculated with 6 $\log_{10}$ PFU of each virus: NE/14 ($n = 2$), AL/39 ($n = 3$), Anhui/1 ($n = 3$), MI/90 ($n = 3$), CO/137 ($n = 3$), CA/147 ($n = 3$; donors in RDT experiment in this study), WA/239 ($n = 3$, donors in RDT experiment in this study), BC/PHL ($n = 3$; donors in RDT experiment in this study), VN/1203 ($n = 3$). For air collection, awake inoculated ferrets were individually placed in sanitized, ventilated plastic transport containers (23.9 L, solid-walled with a perforated lid)[14]. Air was sampled for 30–60 minutes using either the BC251 (National Institute for Occupational Safety and Health) or SPOT (Handix Scientific/Aerosol Devices) samplers (sampling durations and replicates are listed in Supplementary Table 1). The BC251 sampler collected air at 3.5 L/min and separated aerosols into >4 μm, 1–4 μm, and <1 μm fractions[59]. In this study, the >4 μm fraction was resuspended in 600 μL of PBS supplemented with 0.3% bovine serum albumin (BSA) and the 1–4 μm and <1 μm fractions were resuspended in 300 ul of PBS-BSA and pooled for virus quantification. However, most viral RNA and infectious particles were found in the >4 μm fraction, consistent with prior work[18], and total combined titers are reported. The SPOT sampler operated at 1.5 L/min and collected air into 450 uL of PBS with 0.3% BSA kept on ice during collection (sampler settings: conditioner 4–5 °C, initiator 39–40 °C, moderator 21 °C, nozzle 28 °C). Following collection with either of the samplers, each air sample was split: 140 μL was inactivated in AVL buffer (Qiagen) and stored at −80 °C for RNA quantification and the remaining volume was processed immediately by plaque assay to quantify infectious virus[18]. All titers were normalized to 105 L of air to account for differences in sampling time and flow rates. All sampling equipment and containers were disinfected with 70% ethanol, rinsed with water and air-dried after use.

## Virus Quantification

For the detection and quantification of viral RNA copy numbers, total RNA was extracted from the samples using the RNeasy Mini Kit (Qiagen), with 140 μL of sample used for each extraction. Real time RT-PCR was conducted in duplicate using the SuperScript III Platinum One-Step qRT-PCR System (Invitrogen), employing a primer and probe set specific to the influenza A virus M1 gene[18]. The RNA copy numbers for the influenza virus M gene were determined by extrapolating from a standard curve created with samples of known M gene copy numbers. Infectious virus quantification was conducted using a standard plaque assay in MDCK cells[55]. After removal of 140 uL of sample for RNA analyses, the entire remaining volume of each air sample was plated onto multiple wells on a 6-well plate within 1–4 h of collection to prevent freezing (samples with suspected higher viral loads were diluted 10-fold). Titers for nasal washes, rectal swabs, homogenized tissues, and whole blood samples were determined using previously frozen samples. The limits of detection were as follows: 2.9 $\log_{10}$ RNA copies/mL of nasal wash, 2.5 $\log_{10}$ RNA copies/105 L of air, 10 PFU/mL or g of nasal wash or tissue, 1 PFU/105 L of air.

## Hemagglutination inhibition assay

Blood was collected from surviving contact animals on day 21 post-challenge using a BD Vacutainer Rapid Serum Tube (Becton, Dickinson and Company) to assess seroconversion. Briefly, receptor-destroying enzyme (RDE) (Denka Seiken) treated and 10-fold prediluted sera were 2-fold serially diluted and incubated with 8 HAU of virus for 30 minutes before the addition of 1% (vol/vol) horse red blood cells[60]. HAI titers were expressed as the reciprocal of the highest dilution of serum that inhibited hemagglutination.

## Next generation sequencing

Viral RNA was isolated from nasal wash samples using the QIAamp Viral RNA Mini Kit (Qiagen). Portions of the tissues collected from ferrets inoculated with a representative strain from each genotype, CO/137 ($n = 3$) and WA/239 ($n = 3$), were submerged in RNAprotect Tissue Reagent (Qiagen) and stored at 4 °C. The tissues were weighed, placed in buffer RLT Plus (Qiagen), homogenized using an Omni Tissue Homogenizer (Omni International, Inc.), clarified by centrifugation, and viral RNA was isolated using the RNeasy Plus Mini Kit (Qiagen). Whole-influenza-genome amplification was performed using universal Influenza A primers with SuperScript III One-Step RT-PCR with Platinum Taq high-fidelity polymerase kit (Invitrogen) following manufacturer's instructions at half volume reaction size[61]. NGS libraries were prepared with Illumina Nextera DNA Flex kits following manufacturers instructions at half volume reaction size. Libraries were sequenced on and Illumina iSeq100 with paired-end 150-bp reads and v2 chemistry reagents. The Fastq files were trimmed using BBDuk and then assembled to the relevant reference genomes with Bowtie2[62] as implemented in Geneious Prime v2020.2.4, used an CO/137 and WA/239 stock virus sequences in which gene segments were concatenated, separated by a 50 bp "N" spacer. Variants were called with 100x sequence coverage and 5% frequency threshold. Sequencing data have been deposited in the NCBI Sequence Read Archive under BioProject ID: PRJNA1338708.

## Statistical analyses and data presentation

All statistical analyses were performed on measurements taken from distinct samples in R v4.4.0[63]. Area under the curve (AUC) values were calculated using infectious virus and viral RNA copy titers on days 1-3 using the DescTools v0.99.57 package[64]. Statistical assumptions for normality and homogeneity of variances were checked using the Shapiro-Wilk and Levene's test respectively. Statistical group comparisons were performed using the non-parametric tests of Wilcoxon Rank Sum or Kruskal-Wallis with Dunn's multiple comparisons (rstatix v 0.7.2[65]) using Holm's method for p-value adjustment, as specified in the results section. Correlations were performed using Pearson's method with 95% confidence intervals. Pairwise comparisons were conducted using the Kruskal-Wallis test with Dunn's multiple comparisons post hoc analysis. AUC data for infectious virus and RNA was used separately to perform logistic regression modeling for a binary

transmission outcome with 95% confidence intervals. A linear mixed-effects model analyses were performed using data restricted to days 1–3 and structured with one record per ferret per day. Models were fit separately for infectious virus and genomic RNA. Mixed models were fit using lme4 v1.1-35.5[66] with restricted maximum likelihood (REML), the bobyqa optimizer, and an increased iteration limit. We assessed model stability and diagnostics with lme4::isSingular and performance v0.13.0[67]. Day-specific slopes of NW on air were obtained from emmeans v1.10.4[68], and differences between day-specific slopes were evaluated with Tukey-adjusted pairwise comparisons. Degrees of freedom and tests for fixed effects and emmeans estimates used Kenward–Roger approximations provided by lmerTest v3.1-3[69]. Figures were created in R using tidyverse v2.0.0[70], ggplot2 v3.5.1[71], gpubr v0.6.0[72] and patchwork v1.3.0[73] or GraphPad Prism v10.5.0.

### Reporting summary

Further information on research design is available in the Nature Portfolio Reporting Summary linked to this article.

### Data availability

The source data generated in this study have been deposited in GitHub (https://github.com/Troy-Kieran/comparative-airborne-virus-shedding-with-diverse-influenza-strains-in-ferrets)[74]. The Source_Data folder contains the source data file (Figure_1_S1_S2_source_data.xlsx) used to generate Fig. 1, Supplementary Fig. 1, and Supplementary Fig. 2 using GraphPad Prism v10.5.0. and the source data file (FullData.csv) used to generate the remaining figures and to perform statistical analyses using R v4.4.0. Next Generation Sequencing data have been deposited in the NCBI Sequence Read Archive under BioProject ID: PRJNA1338708.

### Code availability

Code used for statistical analyses and figures have been deposited on GitHub https://github.com/Troy-Kieran/comparative-airborne-virus-shedding-with-diverse-influenza-strains-in-ferrets[74]; AR_analysis.R file.

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

## Acknowledgements

We thank the Comparative Medicine Branch at the Centers for Disease Control and Prevention for excellent care of the animals used in this study. Our gratitude goes to Theresa Murray and Brian Hiatt at Washington State Department of Health, Denise Lopez at California Department of Public Health, Shannon Matzinger at Colorado Department of Public Health and Environment, and Nathalie Bastien at the National Microbiology Laboratory at the Public Health Agency of Canada for access to viruses. We would like to thank William Lindsley from National Institute for Occupational Safety and Health for access to BC251

samplers. The findings and conclusions in this report are those of the authors and do not necessarily represent the official position of the Centers for Disease Control and Prevention.

## Author contributions

J.A.P.-P. conceptualized and performed experiments, processed samples, and analyzed data. J.A.B., N.B., C.P., X.S., T.J.K., and T.R.M. assisted with experiments and sample processing. C.P. performed sequence analyses. T.J.K. analyzed data and prepared graphs using R. H.Z. assisted with plaque assays. T.M.T. and T.R.M. supervised the overall work. J.A.C. and Y.H. isolated and characterized A(H5N1) viruses under the supervision of H.D. and C.T.D. J.A.P.-P. wrote the initial draft of the manuscript and all authors reviewed and approved the final version.

## Competing interests

The authors declare no competing interests.
