## [Transparent Peer Review file · Nature Communications]

Advancing A(H5N1) influenza risk assessment in ferrets through comparative evaluation of airborne virus shedding patterns.

Corresponding Author: Dr Joanna Pulit-Penaloza

Version 0:

Reviewer comments:

Reviewer #1

(Remarks to the Author)

This study was conducted by an experienced influenza reference laboratory using the ferret model, the gold standard for risk assessment of emerging B3.13 and D1.1 H5N1 viruses. It provides crucial insights into the potential mammalian transmissibility of these newly emerging H5N1 viruses, particularly the D1.1 variant, which had not previously been assessed in ferrets. Importantly, the authors also employed aerosol sampling using two different air collection platforms and expanded their analysis to include additional H5N1, H9N2, H7N9, and pandemic H1N1 2009 strains with known airborne transmissibility profiles. They provided a detailed analysis of air emission profiles from highly transmissible human seasonal influenza viruses, as well as low and non-transmissible influenza viruses isolated from human cases. The results show that although none of the H5N1 strains transmitted via the air, B3.13 viruses were detected at significantly higher levels compared to D1.1 strains. Strong correlations were observed between viral loads in nasal washes, airborne virus shedding, and transmissibility potential.

Main Concerns:

1. In this study, ferrets were inoculated intranasally with a high dose of 10^6 PFU. While we appreciate that this ensures all ferrets are infected, virus shedding in experimentally infected ferrets may differ from that in real-world settings. For example, in Figure 1, the viral shedding profile in nasal wash samples differed between directly inoculated ferrets and direct-contact ferrets. Since viral replication in directly inoculated ferrets reached peak titers on days 1 and 2, corresponding to higher virus emission in the air, this may not reflect the situation in animals infected with a lower dose. It would be interesting to collect air samples from ferrets infected via direct contact. These ferrets may exhibit different immune response dynamics compared to donor ferrets, which could affect the RNA/PFU ratio, peak viral load, duration, and total amount of virus emitted into the air, as well as onward transmission. If additional animal experiments are not feasible, the authors should discuss these limitations in the Discussion.
2. In the B.3.13 study, significant amounts of infectious virus and viral RNA were detected in the air from directly inoculated B3.13 ferrets. However, no airborne transmission occurred in respiratory contact ferrets. Compared to highly transmissible influenza viruses, low and non-transmissible influenza viruses may have a much higher ID_{50} , which should be taken into account. While determining the ID_{50} would require many animals and conflict with the 3Rs principles, this point should be discussed in the Discussion section.
3. The area under the curve (AUC) of nasal wash and air samples was used to calculate the correlation between influenza viral loads in nasal wash samples and air samples. The authors also compared viral loads from individual days between nasal wash and air samples. It would be helpful to analyse the data using a repeated measures model with data from days 1–3 individually, since the same animals underwent daily nasal wash sampling, which creates multiple repeated measurements within an individual animal.
4. Did the authors sequence the viruses from the nasal wash samples and those collected by the air samplers? Are there any genetic differences?

Others:

The comparison between the two air samplers is very useful and should be published as a reference for other studies using air samplers, such as surveillance and animal studies. For the comparison, data were normalized to facilitate comparison between the two samplers. However, if raw data are compared directly, is the NIOSH sampler still better than SPOT?

Lines 313–314: Can the authors add the temperature data? Did virus emission occur before or after fever?

Reviewer #2

(Remarks to the Author)

In the proposed manuscript, the authors compared the pathogenesis and airborne virus shedding of the clade 2.3.4.4b H5N1 virus genotypes D3.13 and D1.1 in ferrets. These two virus genotypes have recently been found to cause human infections and are widely circulating in wild birds and non-human mammals, raising major concerns about their pandemic potential. In addition to assessing and comparing the pathogenesis and viral shedding with other less and more transmissible influenza viruses, the authors compared the efficiency of two aerosol collectors. Finally, the authors explored the correlation between airborne virus transmission and infectious virus titers and RNA levels detected in air samples.

While the study is well-designed with a strong selection of controls, the manuscript drastically lacks clarity in both the text and data representation. Notably, important data, collected by the authors (rectal swabs, body weight, clinical scoring (e.g. nasal discharge), temperature) need to be presented, especially given the severity of disease caused by the H5N1 viruses. The Materials and Methods section is lacking key experimental details to be able to reproduce the study by others, for example how viral RNA was quantified, how and when the HI's were done on ferret sera, how was virus cultured from blood samples (was any pre-treatment of samples needed? Was whole blood or serum/plasma used, and on which days) etc.

The results section is difficult to navigate, and some conclusions regarding the overall generalization of D1.1 and D3.13 airborne transmission need to be toned down. Notably, some parts of the results or discussion contradict each other. Line 111/113 (results section): "Notably, earlier B3.13 viruses were capable of limited airborne transmission, suggesting both intra- and inter-genotypic variability in transmission phenotypes", line 226/227 (discussion section): "B3.13 viruses are likely not evolving toward a more transmissible phenotype", line 244/246 (discussion section): "This suggests a higher estimated likelihood of airborne transmission for B3.13 viruses compared to D1.1 and indicates that their potential for airborne spread should not be dismissed, underscoring the need for continued monitoring of viruses within this genotype". This final statement is confusing, as no transmission was observed between ferrets with either H5N1 genotype in the presented studies (or in humans). Therefore, it is unclear how the B3.13 viruses can be described as having a higher transmission potential when no transmission was actually detected.

The representation of data is complex and, therefore difficult to interpret. For example, Figure 1 does not clearly indicate the relationship between ferret pairs. Please combine donor-recipient pairs in individual panels for each virus. In Figure 1, panel E, it is very difficult to understand which ferrets the data come from. Are these new ferrets from which only tissue samples were collected, without nasal washes, temperature measurements, or weight loss data? The ferret's group design needs to be better explained and made more easily identifiable in the manuscript. Please also increase the size of the asterisks and 'd4' labels.

The correlation between influenza virus loads in nasal washes and air samples was only done for the first three days. Why was day 5 not included? The data is available for almost all animals.

Line 212/213: How is transmission defined? Usually, it is confirmed by the presence of virus replication in the nasal washes, but here it is based on 'signs of infection'. And what about the other ferret - I don't think it was the same ferret - that seroconverted?

One of the key investigations in the manuscript is the comparison of air sampler efficiency. However, it is surprising that this point is not presented in the title, abstract, or even introduction. We recommend that the authors emphasize their air sampler comparison earlier, rather than only in the results section. Additionally, we advise the authors to present the air sampler comparison in a dedicated paragraph rather than in the 'Airborne virus shedding by influenza virus-infected ferrets' section, as this would greatly improve clarity.

Finally, this study is similar to previous work done by the authors (ref 36). While this manuscript presents novelty in the virus genotypes used and the utilization of the SPOT air sampler in addition to the BC251 air sampler, we advise the authors to clearly place this work in perspective with their previous findings.

Reviewer #3

(Remarks to the Author)

Reviewer #4

(Remarks to the Author)

In this manuscript, Pulit-Penalzo et al. used the ferret model to evaluate the pathogenicity, transmission, and aerosol shedding of human-derived A(H5N1) viruses from two genotypes (B3.13 and D1.1). They also applied aerosol sampling and included additional influenza strains (A(H5N1), A(H9N2), A(H1N1)pdm09) with known transmissibility profiles to

improve methods for pandemic risk assessment.
Overall, this is a well-executed study and a clearly written manuscript.

However, a few minor issues need to be addressed that will improve the manuscript.

1. While the nomenclature is overall correct, the presentation of the four isolates could be made clearer by consistently incorporating their genotype information throughout the text, which would help readers follow the experimental comparisons more easily.
2. Separating Figure 1A–D to align more closely with the narrative in the Results section would make the text easier to follow, as the current layout is somewhat difficult to track while reading.
3. Providing the abbreviations 'DCT' and 'RDT' in the Results section corresponding to Figure 1A–D would improve clarity for the reader.
4. Consider using the same visualization style as in Figure 3, where virus isolates are stratified by transmissibility, to make Figure 2 easier to interpret.
5. It seems like that Figure 5 is mislabeled. The authors refer to Figure 5C, which does not exist.

Version 1:

Reviewer comments:

Reviewer #1

(Remarks to the Author)

The authors have addressed all issues satisfactorily

(Remarks on code availability)

Reviewer #2

(Remarks to the Author)

I'm very pleased with the authors' responses to the comments and think that the manuscript has significantly improved compared to the initial submission. I only have some minor issues:

Line 422: remove space after viruses.

Table 1: The superscript associated with column 4 (named 'Exposure source') should be a not b.
The superscript associated with columns 5 and 6 (named 'DCT' and 'RDT') should be b not c.

(Remarks on code availability)

Reviewer #3

(Remarks to the Author)

(Remarks on code availability)

Reviewer #4

(Remarks to the Author)

The authors have addressed my earlier concerns, and I find their revisions and responses satisfactory.

(Remarks on code availability)

Thank you for your valuable critiques and input; the authors sincerely appreciate the feedback. We are pleased to receive many positive comments, particularly regarding the significance of this work and the interest in our air sampling methods and findings. We agree that all the points raised are important considerations to better understand the multifactorial nature of transmission. We have responded to each comment point by point and made extensive additions to the manuscript as detailed in the responses below. The line numbers refer to specific locations in the clean copy of the resubmitted manuscript. A copy of the manuscript containing tracked changes is also available for review. We believe these changes have significantly enhanced the quality and readability of the manuscript. We hope these improvements meet the reviewers' expectations and that the manuscript is now suitable for publication in Nature Communications. Thank you once again for your time and expert input.

REVIEWER COMMENTS

Reviewer #1 (Remarks to the Author):

This study was conducted by an experienced influenza reference laboratory using the ferret model, the gold standard for risk assessment of emerging B3.13 and D1.1 H5N1 viruses. It provides crucial insights into the potential mammalian transmissibility of these newly emerging H5N1 viruses, particularly the D1.1 variant, which had not previously been assessed in ferrets. Importantly, the authors also employed aerosol sampling using two different air collection platforms and expanded their analysis to include additional H5N1, H9N2, H7N9, and pandemic H1N1 2009 strains with known airborne transmissibility profiles. They provided a detailed analysis of air emission profiles from highly transmissible human seasonal influenza viruses, as well as low and non-transmissible influenza viruses isolated from human cases. The results show that although none of the H5N1 strains transmitted via the air, B3.13 viruses were detected at significantly higher levels compared to D1.1 strains. Strong correlations were observed between viral loads in nasal washes, airborne virus shedding, and transmissibility potential.

Main Concerns:

1. In this study, ferrets were inoculated intranasally with a high dose of 10^6 PFU. While we appreciate that this ensures all ferrets are infected, virus shedding in experimentally infected ferrets may differ from that in real-world settings. For example, in Figure 1, the viral shedding profile in nasal wash samples differed between directly inoculated ferrets and direct-contact ferrets. Since viral replication in directly inoculated ferrets reached peak titers on days 1 and 2, corresponding to higher virus emission in the air, this may not reflect the situation in animals infected with a lower dose. It would be interesting to collect air samples from ferrets infected via direct contact. These ferrets may exhibit different immune response dynamics compared to donor ferrets, which could affect the RNA/PFU ratio, peak viral load, duration, and total amount of virus emitted into the air, as well as onward transmission. If additional animal experiments are not feasible, the authors should discuss these limitations in the Discussion.

Response: One of the rationales for this study was to characterize four new H5N1 viruses isolated from humans and to generate data for risk assessment (IRAT) using protocols that enable comparisons with previous studies. To achieve this, the inoculation dose was set at 10^6 PFU to align with standard protocols. However, we acknowledge the value of conducting further studies with lower, more physiologically relevant doses and/or sampling from contact ferrets to better understand the kinetics and magnitude of strain-specific shedding under varying conditions. Currently, no alternative dose has been recognized as a standard that is considered most physiologically relevant. Ideally, to ensure that all the ferrets are infected

following inoculation with a lower dose, the ferret infectious dose 50 would need to be established for all tested viruses. As the reviewer noted, this would require a substantial number of animals and resources, making it difficult to test a large panel of viruses. To acknowledge these considerations, we have added a paragraph to the discussion (lines 364-370). We also reference studies that enhance our understanding of how dose-dependent ferret inoculations can influence the kinetics and magnitude of shedding. We believe that our comparative analyses using a larger panel of viruses to identify trends in shedding in both nasal washes and air at a fixed dose (ensuring robust infection across strains) provide a solid framework for follow-up studies exploring the feasibility of testing airborne virus shedding following lower inoculation doses.

2. In the B3.13 study, significant amounts of infectious virus and viral RNA were detected in the air from directly inoculated B3.13 ferrets. However, no airborne transmission occurred in respiratory contact ferrets. Compared to highly transmissible influenza viruses, low and non-transmissible influenza viruses may have a much higher ID₅₀, which should be taken into account. While determining the ID₅₀ would require many animals and conflict with the 3Rs principles, this point should be discussed in the Discussion section.

Response: Thank you for your valuable feedback. We agree that this is an important consideration, closely related to the concerns raised in comment 1. We recognize that conducting these experiments would indeed require a large number of animals, which is a significant reason why this area remains understudied in the field. While we were unable to include these additional ferret groups in our study due to the required resources, we referenced studies that support this line of inquiry and highlighted the need for future investigations into the relationship between airborne influenza shedding and the airborne infectious dose 50 in ferrets. We also note that by evaluating ID₅₀ (lines 310-313), this research may provide further insights into the airborne transmissibility of the newly emerging A(H5N1) viruses, particularly regarding the differences between transmissible and non-transmissible B3.13 genotype viruses. We believe that these additions strengthen the manuscript by more clearly discussing the findings and acknowledging the remaining gaps in knowledge.

3. The area under the curve (AUC) of nasal wash and air samples was used to calculate the correlation between influenza viral loads in nasal wash samples and air samples. The authors also compared viral loads from individual days between nasal wash and air samples. It would be helpful to analyse the data using a repeated measures model with data from days 1–3 individually, since the same animals underwent daily nasal wash sampling, which creates multiple repeated measurements within an individual animal.

Response: We acknowledge that there are various methods to analyze this data set. As requested, we utilized linear mixed-effects analyses, which revealed a positive association on day 3 but not on days 1 and 2. Corresponding updates have been made in the Results section (lines 220–225) and the Methods section (lines 488-492). The R code has also been updated on GitHub. Given that this comparative study involves distinct influenza strains and subtypes with significant heterogeneity in host adaptation, frequently reflected in virus shedding kinetics, we consider per-day comparisons to be a weak primary measure. We believe that AUC analysis provides a more robust and reliable assessment of the relationship between nasal wash viral load and airborne virus shedding across diverse viruses.

4. Did the authors sequence the viruses from the nasal wash samples and those collected by the air samplers? Are there any genetic differences?

Response: We performed additional experiments to sequence nasal wash and tissue samples from ferrets infected with representative strains of the B3.13 and D1.1 genotypes (CO/137 and WA/239), to assess

potential viral evolution. The data have been deposited in the NCBI Sequence Read Archive (BioProject ID: PRJNA1338708), and relevant text has been added (lines 116-123). No genetic changes at known mammalian adaptation markers or consistent variant patterns were detected. These results align with previous reports indicating no emergence of mammalian-adaptive variants in ferrets infected with the A/TX/37/2024 2.3.4.4b H5N1 virus. Air samples could not be sequenced due to low viral loads in most samples and workflow limitations. All air samples (the entire volume) were used for plaque assays, leaving no material for further RNA extraction and troubleshooting of low-titer samples by NGS. We have revised the Discussion to acknowledge this limitation and emphasize the need for studies using improved methods to facilitate sequencing of air samples (lines 327-334).

Others:

The comparison between the two air samplers is very useful and should be published as a reference for other studies using air samplers, such as surveillance and animal studies. For the comparison, data were normalized to facilitate comparison between the two samples. However, if raw data are compared directly, is the NIOSH sampler still better than SPOT?

Response: We have expanded the Methods section to provide a clearer description of the procedures used. We carefully considered how to compare the samplers, given that they draw air at different rates. The SPOT sampler draws 1.5 L/min, versus 3.5 L/min for the NIOSH sampler (2.33-fold difference). While normalized data could be expressed as a value per minute or hour, our normalized data are presented as a value per liter of air. To normalize the data, the SPOT sampler values were multiplied by 2.33 to align with those of the NIOSH sampler. Without this normalization, the SPOT sampler would have values 2.33-times lower than those presented here. Consequently, the SPOT sampler collects fewer particles than the NIOSH sampler and has a lower collection efficiency. We have added additional text to clarify this point (lines 191-193).

Lines 313–314: Can the authors add the temperature data? Did virus emission occur before or after fever?

Response: The peak emission occurred concurrently with or after the peak in temperature. We have added Supplementary Figure 1A and 1B to show the weight and temperature data. Additionally, the mean peak temperature values and the median day of peak occurrence are now presented in Supplementary Table 5.

Reviewer #2 (Remarks to the Author):

In the proposed manuscript, the authors compared the pathogenesis and airborne virus shedding of the clade 2.3.4.4b H5N1 virus genotypes D3.13 and D1.1 in ferrets. These two virus genotypes have recently been found to cause human infections and are widely circulating in wild birds and non-human mammals, raising major concerns about their pandemic potential. In addition to assessing and comparing the pathogenesis and viral shedding with other less and more transmissible influenza viruses, the authors compared the efficiency of two aerosol collectors. Finally, the authors explored the correlation between airborne virus transmission and infectious virus titers and RNA levels detected in air samples.

While the study is well-designed with a strong selection of controls, the manuscript drastically lacks clarity in both the text and data representation.

Notably, important data, collected by the authors (rectal swabs, body weight, clinical scoring (e.g. nasal discharge), temperature) need to be presented, especially given the severity of disease caused by the H5N1 viruses.

Response: Thank you for this comment. The clinical signs, including mean maximum temperature (including the median peak day), weight change from the baseline, nasal discharge, diarrhea, and viremia are presented in Table 2. We have added Supplementary Fig. 1A and B to show weight and temperature change kinetics, and we also added Supplementary Fig. 2 A-F to show the time course for rectal swab titers.

The Materials and Methods section is lacking key experimental details to be able to reproduce the study by others, for example how viral RNA was quantified, how and when the HI's were done on ferret sera, how was virus cultured from blood samples (was any pre-treatment of samples needed Was whole blood or serum/plasma used, and on which days) etc.

Response: We have made extensive revisions to the Methods section (with tracked changes) to include more detailed information, and we now provide all the requested information. We are happy to hear that our methods could serve as a valuable resource for others.

The results section is difficult to navigate, and some conclusions regarding the overall generalization of D1.1 and D3.13 airborne transmission need to be toned down. Notably, some parts of the results or discussion contradict each other. Line 111/113 (results section):” Notably, earlier B3.13 viruses were capable of limited airborne transmission, suggesting both intra- and inter-genotypic variability in transmission phenotypes”, line 226/227 (discussion section): ”B3.13 viruses are likely not evolving toward a more transmissible phenotype”, line 244/246 (discussion section): ”This suggests a higher estimated likelihood of airborne transmission for B3.13 viruses compared to D1.1 and indicates that their potential for airborne spread should not be dismissed, underscoring the need for continued monitoring of viruses within this genotype”. This final statement is confusing, as no transmission was observed between ferrets with either H5N1 genotype in the presented studies (or in humans). Therefore, it is unclear how the B3.13 viruses can be described as having a higher transmission potential when no transmission was detected.

Response: Thank you for pointing out these inconsistencies, your feedback is very helpful. We have removed lines 226-227 and 244-246 to avoid confusion and to prevent generalizations about virus evolution, especially given the mixed phenotypes observed. We also toned down our conclusions throughout the manuscript.

The representation of data is complex and, therefore difficult to interpret. For example, Figure 1 does not clearly indicate the relationship between ferret pairs. Please combine donor-recipient pairs in individual panels for each virus. In Figure 1, panel E, it is very difficult to understand which ferrets the data come from. Are these new ferrets from which only tissue samples were collected, without nasal washes, temperature measurements, or weight loss data? The ferret's group design needs to be better explained and made more easily identifiable in the manuscript. Please also increase the size of the asterisks and 'd4' labels.

Response: As requested, we have combined donor-recipient pairs in individual panels for each virus and increased the size of the asterisks and d4 labels in Figure 1. The ferrets in Figure 1 G (previously 1E) represent 6 new ferrets inoculated with CO/137 and WA/239 (for the sole purpose of examining day 3 tissue titers, as performed in previous publications on other H5N1 strains), while the ferrets inoculated with CA/147 and BC/PHL are the same ferrets as the donors in Fig 1C and F (from which tissues were collected

when these ferrets reached humane endpoint criteria). We revised the Figure 1 caption for better readability and updated the color scheme.

The correlation between influenza virus loads in nasal washes and air samples was only done for the first three days. Why was day 5 not included? The data is available for almost all animals.

Response: We limited our analyses to days 1-3 because some virus groups lacked day 5 data, either due to animals meeting humane endpoint criteria or because the data were collected on day 4 instead of day 5. Including incomplete day 5 data could skew the analyses and lead to inaccurate conclusions. We have added text to acknowledge this experimental choice in lines 209, 359-364.

Line 212/213: How is transmission defined? Usually, it is confirmed by the presence of virus replication in the nasal washes, but here it is based on ‘signs of infection’. And what about the other ferret - I don’t think it was the same ferret - that seroconverted?

Response: Thank you for your feedback. We have clarified the definition of transmission throughout the study, emphasizing that it is not based on signs of infection. In this study, transmission is defined by virus detection in any specimen collected and/or seroconversion. We updated the Methods section (lines 413-314) and the caption of Table 1 for clarification, and we further refined the Discussion to better describe which ferrets seroconverted (lines 275-279). One WA/239 ferret had virus detected in a rectal swab sample and, due to severe illness, required euthanasia; we subsequently collected tissues and detected infectious virus in the lower respiratory tract. The second ferret seroconverted without viral detection and survived, so we were unable to test its tissues.

One of the key investigations in the manuscript is the comparison of air sampler efficiency. However, it is surprising that this point is not presented in the title, abstract, or even introduction. We recommend that the authors emphasize their air sampler comparison earlier, rather than only in the results section. Additionally, we advise the authors to present the air sampler comparison in a dedicated paragraph rather than in the ‘Airborne virus shedding by influenza virus-infected ferrets’ section, as this would greatly improve clarity. Needs revision of text and maybe supp fig.

Response: Thank you for your valuable suggestion. We agree that it is important to present the sampler comparison more effectively. Due to word limit, we were unable to include additional detail in the title, and for the same reason, we could only make minor adjustments in the abstract (we now specify the types of samplers in the abstract). We have significantly expanded the introduction (lines 56-77) and revised the section titled “Airborne Virus Shedding by Influenza Virus-Infected Ferrets” to “Comparative Evaluation of Airborne Virus Shedding Using BC251 and SPOT Samplers,” dedicating that section to a detailed comparison of the samplers. We have two figures and one table focused on sampler comparisons: Figure 3, Supplementary Figure 3, Supplementary Table 2.

Finally, this study is similar to previous work done by the authors (ref 36). While this manuscript presents novelty in the virus genotypes used and the utilization of the SPOT air sampler in addition to the BC251 air sampler, we advise the authors to clearly place this work in perspective with their previous findings.

Response: We have added an additional paragraph to the Introduction to clarify how this current research builds on previous findings (lines 56-77, 202-207) and to provide more context regarding the importance and challenges of aerosol sampling, as requested in comment 7.

Reviewer #3 (Remarks to the Author):

Reviewer #4 (Remarks to the Author):

In this manuscript, Pulit-Penaloza et al. used the ferret model to evaluate the pathogenicity, transmission, and aerosol shedding of human-derived A(H5N1) viruses from two genotypes (B3.13 and D1.1). They also applied aerosol sampling and included additional influenza strains (A(H5N1), A(H9N2), A(H1N1)pdm09) with known transmissibility profiles to improve methods for pandemic risk assessment.

Overall, this is a well-executed study and a clearly written manuscript.

However, a few minor issues need to be addressed that will improve the manuscript.

1. While the nomenclature is overall correct, the presentation of the four isolates could be made clearer by consistently incorporating their genotype information throughout the text, which would help readers follow the experimental comparisons more easily.

Response: We made additions throughout the text to more consistently incorporate their genotype information.

2. Separating Figure 1A–D to align more closely with the narrative in the Results section would make the text easier to follow, as the current layout is somewhat difficult to track while reading. ?

Response: We have edited the figure to separate the data on direct contact and respiratory transmission for improved clarity. Additionally, we updated the color scheme to facilitate tracking of individual ferrets. We also made edits to the results text and the figure caption for better readability.

3. Providing the abbreviations ‘DCT’ and ‘RDT’ in the Results section corresponding to Figure 1A–D would improve clarity for the reader.

Response: We clarified the abbreviations and provided additional background information to enhance readability. Lines 126-128, 147-150.

4. Consider using the same visualization style as in Figure 3, where virus isolates are stratified by transmissibility, to make Figure 2 easier to interpret.

Response: We have made changes to Figure 2 as suggested. The viruses are now ordered and grouped by their transmissibility profiles to ensure that the figure formatting aligns exactly with that of Figure 3.

5. It seems like that Figure 5 is mislabeled. The authors refer to Figure 5C, which does not exist.

Response: Thank you for bringing this error to our attention. We have corrected the figure labels.

Thank you for re-reviewing the manuscript. We are pleased that the changes we made are satisfactory.

REVIEWER COMMENTS

Reviewer #1 (Remarks to the Author):

The authors have addressed all issues satisfactorily

Response: Thank you.

Reviewer #2 (Remarks to the Author):

I'm very pleased with the authors' responses to the comments and think that the manuscript has significantly improved compared to the initial submission. I only have some minor issues:

Line 422: remove space after viruses.

Table 1: The superscript associated with column 4 (named 'Exposure source') should be a not b.

The superscript associated with columns 5 and 6 (named 'DCT' and 'RDT') should be b not c.

Response: All typos were corrected as requested. Thank you.

Reviewer #3 (Remarks to the Author):

Response: Thank you.

Reviewer #4 (Remarks to the Author):

The authors have addressed my earlier concerns, and I find their revisions and responses satisfactory.

Response: Thank you.